# pH-Sensitive Peptide Hydrogels as a Combination Drug Delivery System for Cancer Treatment

**DOI:** 10.3390/pharmaceutics14030652

**Published:** 2022-03-16

**Authors:** Yuanfen Liu, Yingchun Ran, Yu Ge, Faisal Raza, Shasha Li, Hajra Zafar, Yiqun Wu, Ana Cláudia Paiva-Santos, Chenyang Yu, Meng Sun, Ying Zhu, Fei Li

**Affiliations:** 1Department of Pharmacy, Jiangsu Health Vocational College, Nanjing 211800, China; liuyuanfen2010@gmail.com; 2National Clinical Research Center for Child Health and Disorders, Ministry of Education Key Laboratory of Child Development and Disorders, Department of Emergency, Children’s Hospital of Chongqing Medical University, Chongqing 400014, China; tongtong20100@163.com; 3State Key Laboratory of Natural Medicines, China Pharmaceutical University, Nanjing 210009, China; 18851109616@163.com (Y.G.); wuyiquncpu@163.com (Y.W.); yuchenyang0929@163.com (C.Y.); 2020170314@stu.cpu.edu.cn (M.S.); 4School of Pharmacy, Shanghai Jiao Tong University, 800 Dongchuan Road, Shanghai 200240, China; hajrazafar@sjtu.edu.cn; 5College of Pharmacy, Xinjiang Medical University, Ürümqi 830000, China; lishaf@126.com; 6Department of Pharmaceutical Technology, Faculty of Pharmacy, University of Coimbra, 3000-548 Coimbra, Portugal; acsantos@ff.uc.pt

**Keywords:** combination therapy, drug delivery, hydrogel, cancer, peptide

## Abstract

Conventional antitumor chemotherapeutics generally have shortcomings in terms of dissolubility, selectivity and drug action time, and it has been difficult to achieve high antitumor efficacy with single-drug therapy. At present, combination therapy with two or more drugs is widely used in the treatment of cancer, but a shortcoming is that the drugs do not reach the target at the same time, resulting in a reduction in efficacy. Therefore, it is necessary to design a carrier that can release two drugs at the same site. We designed an injectable pH-responsive OE peptide hydrogel as a carrier material for the antitumor drugs gemcitabine (GEM) and paclitaxel (PTX) that can release drugs at the tumor site simultaneously to achieve the antitumor effect. After determining the optimal gelation concentration of the OE polypeptide, we conducted an in vitro release study to prove its pH sensitivity. The release of PTX from the OE hydrogel in the medium at pH 5.8 and pH 7.4 was 96.90% and 38.98% in 7 days. The release of GEM from the OE hydrogel in media with pH of 5.8 and 7.4 was 99.99% and 99.63% in 3 days. Transmission electron microscopy (TEM) and circular dichroism (CD) experiments were used to observe the microstructure of the peptides. The circular dichroism of OE showed a single negative peak shape when under neutral conditions, indicating a β-folded structure, while under acidic conditions, it presented characteristics of a random coil. Rheological experiments were used to investigate the mechanical strength of this peptide hydrogel. Furthermore, the treatment effect of the drug-loaded peptide hydrogel was demonstrated through in vitro and in vivo experiments. The results show that the peptide hydrogels have different structures at different pH values and are highly sensitive to pH. They can reach the tumor site by injection and are induced by the tumor microenvironment to release antitumor drugs slowly and continuously. This biologically functional material has a promising future in drug delivery for combination drugs.

## 1. Introduction

Cancer has become one of the leading causes of mortality in the world. Single-drug therapy has difficulty achieving high efficacy [1]. As a result, combination drug delivery has been widely used in cancer treatment. To enhance the therapeutic effect of combination drugs, two drugs can be co-loaded and delivered to tumor tissues to achieve synergistic drug delivery [2,3]. GEM is a water-soluble difluorinated nucleoside antimetabolite drug. It is remarkably effective for a wide range of solid tumors when used alone, but its toxicity is relatively high due to its short half-life and the need for continuous high doses [4]. PTX is a strongly lipid-soluble diterpene alkaloid drug. It is less toxic than some other antitumor therapies, but it has other problems, such as poor bioavailability [5]. Therefore, to reduce side effects and improve efficacy, PTX in combination with GEM is often used for cancer treatment, such as breast cancer, pancreatic cancer and bladder cancer. However, this drug combination still has several problems. The most remarkable one is that the drugs cannot be delivered to the target location at the same time, which prevents the maximum efficacy from being realized.

In recent years, the main two-drug co-delivery systems reported in the literature have been liposomes [6], nanoparticles [7], drug-carrier copolymers [8] and hydrogels [9]. A hydrogel is a three-dimensional network system formed by cross-linked polymers or fibers in an extended fluid. It can swell in water [10]. Because of its good water absorption and reversibility, hydrogels have been widely used in biomedical applications, such as drug [11] or cell delivery [12] and tissue engineering [13]. Responsive hydrogels are new types of hydrogels that undergo structural changes in response to external physical or chemical stimuli [14]. They have the advantages of good biocompatibility and responses to external environmental conditions. They can release drugs in the designated area at the same time, achieving the goal of targeted drug delivery. According to different external stimuli, responsive hydrogels are mainly classified as temperature-sensitive [15], pH-sensitive [16], light-sensitive [17], multi-sensitive [18] and electrical stimuli-responsive [19]. In addition, some release mechanisms are dose-dependent [20].

Currently, most polymer hydrogel materials have poor biodegradability when used as drug delivery carrier materials, which may be harmful to humans [21]. In contrast, peptide hydrogels offer significant advantages of good biocompatibility, controlled degradability, lower immunogenicity and higher bioavailability. They are easy to prepare and modify and have great advantages in drug delivery. However, some peptide hydrogels have low pH sensitivity in the tumor microenvironment. After being injected into the body, they leak a large amount of the drug before forming the hydrogel. They also have the shortcomings of poor mechanical strength and low bioavailability [22].

To address these issues, we prepared an injectable pH-responsive peptide hydrogel encapsulating GEM and PTX as a two-drug co-delivery system, which we named OE (VKVKVOVK-V^D^PPT-KVEVKVKV-NH_2_). The sequence is composed of alternate polar and non-polar amino acids, including valine and -V^D^PPT-. At the pH value under physiological conditions, the polypeptide can form nanofibers through the self-assembly of β-sheet secondary structures. When the polypeptide concentration reaches a certain level, the nanofibers in the polypeptide structure cross-link through self-assembly to form a three-dimensional network structure, which can encapsulate active drugs. In a slightly acidic tumor microenvironment, the three-dimensional network is disrupted, and the encapsulated drugs are released, thereby inhibiting tumor growth. This system can continuously release active drugs in a safe and effective dose at the tumor site, which improves the therapeutic effect and reduces side effects of the drugs. In this study, the OE peptide was designed and synthesized by a solid-phase peptide synthesis method. The system was able to form peptide hydrogels encapsulating GEM and PTX in vitro with a suitable mechanical strength under neutral conditions. In the process of drug release, the hydrophilic drug GEM is first released in large quantities and can be completely released in the first three days, while the hydrophobic drug PTX continues to be released slowly until it is completely released on the seventh day. This not only prolongs the duration of action of the drug but also enhances the therapeutic effect of the drug and realizes an anti-breast cancer effect [23]. In addition, through in vitro and in vivo studies on its antitumor activity, we further successfully confirmed that this system has a strong inhibitory effect on tumor growth and metastasis, which enhances the therapeutic effect of the drugs. In addition, we also confirmed that the blank peptide hydrogel in the system is safe and non-toxic and has good development prospects.

## 2. Material and Methods

### 2.1. Materials

Peptides were purchased from Peptide Synthesis Company. GEM and PTX were received from the National Institute for Food and Drugs (Shanghai, China). Unless otherwise specified, all other reagents were obtained from Sigma-Aldrich (St. Louis, MO, USA).

### 2.2. Methods

#### 2.2.1. Design, Synthesis, Separation and Purification of the Peptide

First, based on MAX1 (VKVKVKVK–V^D^PPT–KVKVKV–NH_2_) [24], we designed a peptide sequence with properties more suitable for the delivery of anticancer drugs. According to previous experiments [25], we considered replacing lysine (K) with glutamic acid (E) in the MAX1 sequence, which would reduce the charge of the peptide, increase the mechanical strength of the gel formed by the peptide and improve the stability of the drug-loaded gel. The isoelectric point of ornithine (O) is 10.80 and that of lysine (K) is 9.74. We replaced K in the MAX1 sequence with O to make the amino acid side chain more sensitive to protonation as well as pH. Through the previous investigation, it was discovered that the 15th amino acid in MAX1 had a significant impact on the properties of the peptide [26]. In addition, previous research [27] showed that the substitution of K at position 6 in the MAX1 sequence by O could improve the acid response. Finally, a new peptide sequence based on MAX1 was designed as follows. Ornithine was used to substitute the amino acid at position 6, and glutamic acid was used to substitute the amino acid at position 15. The complete peptide sequence of this new peptide was (VKVKVOVK-V^D^PPT-KVEKVKV-NH_2_), and we named it OE. The Fmoc orthogonal protection strategy was adopted in the solid-phase peptide synthesis method, and a microwave synthesizer (CEM) was used to synthesize OE peptides [28,29]. We employed RHPLC (SHIMADZU, HPLC LC-10ATVP, Tokyo, Japan) to separate and purify the OE peptide. After purification, a liquid-phase mass spectrometer (LC-MS, ACQUITY UPLC H-Class-XEVO TQD, Waters, Milford, MA, USA) was used to characterize the product obtained by preparing the liquid phase.

#### 2.2.2. Preparation of GEM+PTX-Loaded Peptide Hydrogels

In order to prepare a GEM+PTX suspension, paclitaxel and gemcitabine with a mass ratio of 1:4 [30] were dissolved in a 150 mM sodium chloride solution. The OE peptide was dissolved in the above suspension, and after it was entirely dissolved, we adjusted the pH of the solution to 7.4. After standing for a period of time, the drug-loaded hydrogel was more stable.

#### 2.2.3. Investigation of the Gelation Factors of OE Peptide

The effects of OE concentration and solution pH on the gelation of OE polypeptide were investigated. We prepared OE polypeptide hydrogels with concentrations of 10, 15 and 20 mg/mL and observed their gel formation under neutral conditions. After that, we prepared 15 mg/mL OE polypeptide hydrogel and observed its gel formation under different pH conditions.

#### 2.2.4. pH Sensitivity of the OE Peptide Hydrogel

We conducted an in vitro release study on the OE polypeptide hydrogel to confirm its pH responsiveness. A 15 mg/mL OE polypeptide hydrogel was prepared according to the above gel preparation method, and 0.25 mg/mL GEM and 1 mg/mL PTX were loaded into the OE hydrogel. The stable drug-loaded hydrogel was centrifuged at 1000 rpm for 4 min to remove air bubbles and stored in a refrigerator overnight. We washed the surface of the hydrogel twice with pH 7.4 phosphate buffer. Subsequently, we added 1 mL of pH 7.4 phosphate buffer or 1 mL of pH 5.8 phosphate buffer to the OE-GEM-PTX hydrogel. In addition, 0.5% Tween 80 was added to the buffer to increase the solubility of paclitaxel. At 0.5, 1, 2, 6, 12, 24, 48, 72, 96, 120 and 168 h, all release solutions were aspirated to determine the contents of GEM and PTX. At the same time, we added 1 mL of fresh release fluid. The contents of GEM and PTX were measured by HPLC. The conditions for determining the content of GEM were as follows: column: Kromasil 100-5C18 (4.6 × 250 mm, 5 μm); detection wavelength: 268 nm; column temperature: 40 °C; phase A (water) flow rate: 0.95 mL/min; phase B (acetonitrile) flow rate: 0.05 mL/min; volume: 20 μL. The conditions for determining the content of PTX were as follows: column: Kromasil 100-5C18 (4.6 × 250 mm, 5 μm); detection wavelength: 227 nm; column temperature: 40 °C; phase A (water) flow rate: 0.52 mL/min; phase B (acetonitrile) flow rate: 0.48 mL/min; volume: 20 μL.

#### 2.2.5. Secondary Structure

In order to investigate the secondary structure of peptide molecules in different conditions, we employed circular dichroism (CD) detection. A Jasco J-810 spectropolarimeter was used to gather the far-ultraviolet CD spectrum of OE peptide hydrogels. The pH 7.4 and pH 5.8 buffers were used to dissolve OE polypeptides to prepare hydrogels, and 100 μL of each solution was placed in a 0.1 mm quartz cell. The parameter settings were as follows: wavelength range: 190–260 nm; bandwidth: 1 nm; response time: 1 s; scanning speed: 50 nm/min.

#### 2.2.6. Transmission Electron Microscopy (TEM)

Transmission electron microscope images (TEM) were acquired to observe the microstructure of the peptide under different conditions. Peptide hydrogels with pH 7.4 and pH 5.8 were prepared. A drop of the diluted solution was taken and placed evenly on the carbon-coated copper mesh. We used filter paper to absorb the excess liquid after 3 min. Then, we used 2% phosphotungstic acid solution for negative staining for 3–5 min. After drying under infrared light, a TEM (Hitachi, HT-7700, Tokyo, Japan) was used to observe the microstructure of the OE peptide sample.

#### 2.2.7. Rheological Study

The peptide hydrogel used for antitumor drug delivery requires high mechanical strength to prevent its collapse at the tumor site and prevent release and leakage of the drugs. Since this delivery carrier forms a gel in vitro, it is necessary to deliver it to the tumor site by injection. Therefore, the gel also needs to be diluted for the injection. At the same time, it must quickly reform into a hydrogel with suitable mechanical strength after being injected into the body. In this study, we measured the mechanical strength and injectable properties of the hydrogel carrier by measuring viscoelastic parameters such as the modulus of storage (G′) and loss modulus (G) of polypeptide hydrogels by rotational rheometer.

In order to study the mechanical strength as well as the injectability of the hydrogel system, the viscoelastic parameters were measured by the HAAKE600 rotary rheometer (Japan Jasco Company, Tokyo, Japan). We prepared blank OE peptide hydrogels and OE peptide hydrogels loaded with GEM+PTX. Dynamic frequency scanning and dynamic time scanning were performed on blank OE hydrogel and drug-loaded OE polypeptide hydrogel, respectively. The parameters of the dynamic frequency sweep were set as follows: the frequency was 0.1–100 rad/s, and the shear force was 1%. The parameters of the dynamic time sweep were set as follows: fixed frequency: 1 Hz; shear force: 1% at 0–3 min, 100% at 3–5 min, and 1% at 5–37 min. Then, the above process was repeated at 37–70 min.

#### 2.2.8. In Vitro Cytotoxicity Studies

##### Cell Culture

The mouse breast cancer cell line (4T1) was purchased from the Cell Resource Center of Shanghai Institute of Biological Sciences, Shanghai, China. 4T1 cells were cultured in DMEM medium containing 10% fetal bovine serum and double antibodies in a 5% CO_2_ incubator at 37 °C.

##### Biocompatibility of Blank Peptide Hydrogel

In order to confirm the safety and non-toxicity of the gel carrier, a biocompatibility test of the OE hydrogel was carried out through the Cell Counting Kit-8 (CCK8). The OE polypeptide solutions were prepared with concentrations of 0, 1, 10, 50, 100 and 500 μg/mL, and the pH of the solution was adjusted to form a stable hydrogel. 4T1 cells were cultured in the above-mentioned medium. When the density reached 90%, the cells were digested to obtain a resuspension. In a 96-well plate, 4T1 cells were seeded at a concentration of 5 × 104 cells/mL with 100 microliters per well. After stabilizing for 24 h, the prepared OE polypeptide solution was diluted with culture medium. The original medium was removed from a set of 96-well plates, and fresh medium containing polypeptides was added as a control group. Fresh medium was added to another set of 96-well plates. After 24 h, we added 10 microliters of CCK8 solution to each well as the experimental group. After incubating the cells for 1–2 h, we used a microplate reader to read at 450 nm. Another group containing only medium was used as the blank group. The cell survival ratio was then calculated.

The calculation formula for the cell survival rate of each group was as follows:$$\mathrm{cell}\;\mathrm{viability}\left(\%\right)=\frac{A_{\mathrm{experimental}\;\mathrm{group}}-A_{\mathrm{blank}\;\mathrm{group}}}{A_{\mathrm{control}\;\mathrm{group}}-A_{\mathrm{blank}\;\mathrm{group}}}\times 100\%$$

##### In Vitro Antitumor Efficacy

In this part, we studied the ability of different concentrations of GEM and PTX to inhibit cell proliferation. Solutions of 15 mg/mL OE peptide were prepared. Groups were determined as follows: (1) normal saline at a certain concentration (control group); (2) OE peptide hydrogel with the GEM solutions at concentrations of 0.025, 0.25, 1.25, 2.5 and 12.5 μg/mL (GEM@H group); (3) OE peptide hydrogel with the PTX solutions at concentrations of 0.1, 1, 5, 10 and 50 μg/mL (PTX@H group); (4) free GEM+PTX solution with the same concentrations as the above grouping (GEM+PTX group); (5) OE peptide hydrogel with GEM+PTX with the same concentrations as the above grouping (GEM+PTX@H group). After 4T1 cells were cultured for 48 h, in vitro cytotoxicity was determined by the CCK-8 method [31].

#### 2.2.9. In Vivo Antitumor Research

##### Statement

All animal experiments complied with the ARRIVE guidelines and were carried out in accordance with the U.K. Animals (Scientific Procedures) Act, 1986.

##### Establishment of the Tumor Model

We obtained BALB/c female mice from Nanjing Jilin Biotechnology Development Co., Ltd. (Nanjing, China). Animal experiments were carried out according to the guidelines of the European Community. After disinfecting the skin of the right armpit of female BALB/c mice, they were inoculated with 4T1 mouse breast cancer cell suspension with a concentration of 5 × 107 cells/mL. Guidelines for care and use of laboratory animals of China Pharmaceutical University were used to perform animal studies, and these studies were duly approved by the animal ethics committee of China Pharmaceutical University (no.: 2021-12-018; date: 16 December 2021).

##### In Vivo Drug Efficacy Evaluation

Twenty-five tumor-bearing mice were randomly divided into five groups. The tumors of these mice were about 100 mm^3^ and weighed 16~18 g. The 5 groups of mice were subcutaneously injected as follows: (1) 100 μL/20 g saline (control group); (2) 100 μL/20 g GEM suspension, in which the GEM dose was 10 mg/kg (GEM@H group); (3) 100 μL/20 g PTX suspension, in which the PTX dose was 10 mg/kg (PTX@H group); (4) 100 μL/20 g GEM+PTX suspension, in which the GEM dose was 1.25 mg/kg and the dose of PTX was 5 mg/kg (GEM+PTX group); (5) 100 μL/20 g OE peptide hydrogel loaded with GEM+PTX (GEM+PTX@H group). After 0, 1, 4 and 7 days, we measured the body weight as well as the tumor volume of tumor-bearing mice. The tumor volume was calculated as follows:$$\mathrm{Tumor}\;\mathrm{volume}\left({\mathrm{mm}}^{3}\right)=\frac{\mathrm{Longest}\;\mathrm{tumor}\;\mathrm{diameter}-\mathrm{Shortest}\;\mathrm{tumor}\;{\mathrm{diameter}}^{2}}{2}$$

On the seventh day, all mice were euthanized. Their tumors were removed to measure their weight and volume. TUNEL analysis was also performed on tumor sections from the five groups of mice.

In addition, the hearts, livers, spleens, lungs and kidneys of mice in the control group, GEM+PTX group and GEM+PTX@H group were removed and examined by H&E staining.

##### Biodistribution of the Peptide Hydrogel

The in vivo imaging system can observe the distribution of Cy5 fluorescent probes in tumor-bearing mice, which can be used to investigate the biodistribution of drug-loaded hydrogels. We randomly divided six tumor-bearing mice into two groups, with three mice in each group. These mice had similar body weights and tumor sizes. The 2 groups of mice were subcutaneously injected as follows: (1) 100 μL/20 g Cy5 probe, in which the dose was 1 mg/kg (Cy5 group); (2) 100 μL/20 g Cy5-loaded OE peptide hydrogel, in which the OE dose was 1 mg/kg (OE-Cy5 group). Fluorescence imaging of anesthetized tumor-bearing mice was performed 1, 6, 12, 24, 48 and 96 h after administration. After that, all mice were euthanized, and their hearts, lungs, livers, spleens, kidneys and tumors were removed. We employed an imager to observe the distribution of the Cy5 probes in both groups.

##### Biocompatibility of OE Peptide Hydrogels In Vivo

The in vivo biocompatibility of OE polypeptide hydrogels was investigated to determine whether the materials are safe in vivo. Two groups of nude mice were injected with normal saline or 15 mg/mL blank OE polypeptide hydrogel subcutaneously in their backs. The injection dose was 100 μL/20 g. After 96 h, all mice were euthanized. We removed the subcutaneous tissues around the injection site on the back. In order to observe whether it had caused inflammation, skin necrosis or other adverse reactions, H&E staining was employed.

## 3. Results and Discussion

### 3.1. Preparation of GEM+PTX-Loaded Peptide Hydrogel

The gelation of OE at different concentrations and pH is shown in Table 1 and Table 2.

The results showed that a stable hydrogel could be formed when the concentration of the OE peptide reached a certain level. It was also found that the hydrogel formed more quickly when the concentration of the OE peptide was higher. This is shown in Table 1 and Table 2. The OE peptide formed a stable hydrogel under neutral conditions. Therefore, it is a pH-sensitive carrier material and can be loaded with antitumor drugs.

### 3.2. pH Sensitivity of the OE Peptide Hydrogel

In this part, we performed an in vitro release experiment using the OE peptide. Figure 1A shows the results. The total release of PTX from the OE hydrogel in the medium at pH 5.8 was 96.90% in 7 days, and in the medium at pH 7.4, it was 38.98%. The total release of GEM from the OE hydrogel in the medium at pH 5.8 and pH 7.4 was 99.99% and 99.63% in 3 days. The reason for this result is that GEM is highly hydrophilic and can quickly diffuse into aqueous solutions.

According to the results, the OE hydrogel is responsive to pH. It can continuously release PTX under acidic conditions, and acidic conditions are similar to the tumor microenvironment. Therefore, the OE peptide not only has good gelation properties but also responds in a pH-sensitive manner. The release speed can be controlled by regulating the peptide concentration. The results show that OE has the desired peptide sequence.

### 3.3. Secondary Structure

The circular dichroism results are shown in Figure 1B. The pictures show that when OE is under neutral conditions of pH 7.4, it has a β-folded structure, as indicated by the single negative peak shape, while under acidic conditions of pH 5.8, it presents characteristics of a random coil, that is, negative peaks at short wavelengths.

These results indicate that this peptide sequence can form a stable hydrogel by β-folding under neutral conditions and is destroyed under acidic conditions. This shows that the OE peptide is pH-responsive.

### 3.4. TEM

Figure 1C,D shows the results of electron microscopy of the 15 mg/mL neutral and acidic OE peptide hydrogel.

Comparing Figure 1C,D shows that under neutral conditions, OE forms a slender nanometer fiber network. The illustrated network is suitable for packaging antitumor drugs. Under acid conditions, electron transmission microscopy did not show a nanofibrous peptide. The peptide is shorter and scattered, which is different from its form under neutral conditions. It can be concluded that this peptide has good pH sensitivity.

### 3.5. Rheological Study

The scanning results of OE sequences of blank hydrogels and drug-loading gels are shown in Figure 1E. Figure 1F shows the results after the hydrogel was injected into the specified part by exerting thrust in the syringe. After 3 min of shearing, the shear force suddenly rose to 100% and then returned to 1%. The results of dynamic time scanning showed that when the shear force increased to 100% for 3 min, G′ changed from 135 Pa to 143 Pa, while when the shear force decreased to 1% for 5 min, the G′ of the peptide hydrogel quickly recovered to more than 4000 Pa.

In Figure 1E, there is no significant change in rheological properties, regardless of whether or not the drug is loaded. The value of the storage modulus (G′) is about 10 times larger than that of the loss modulus (G′′), which proves that it is a stable carrier material with strong rigidity and solid-like properties. The results in Figure 1F indicate that this hydrogel can produce a certain degree of fluidity when subjected to shear force and can quickly reform into a stable peptide hydrogel after the shear force is removed. This indicates that the peptide gel can be injected, and thus, the pain caused by surgical implantation can be reduced.

### 3.6. In Vitro Cytotoxicity Studies

Based on the characterization, the cytotoxicity of drug-loaded OE was further studied. The CCK-8 assay was used to investigate the cytotoxicity of blank hydrogels and to explore their biocompatibility. In Figure 2A, the survival rates of 4T1 cells after 48 h of co-incubation with OE polypeptide hydrogels with different concentrations are all above 97%.

The results show that concentrations within this range are safe. The material may be suitable for antitumor drug delivery. Figure 2B shows the suppression of 4T1 cell proliferation by GEM and PTX at different concentrations. As shown in the figure, the cytotoxicity of OE hydrogels with drugs was similar to that of free GEM and PTX. This indicates that GEM and PTX can still exert antitumor activity after being incorporated into peptide hydrogels. Comparing the results of both GEM and PTX encapsulated by hydrogels and the results of the individual drugs encapsulated by hydrogels, it is confirmed that the combined treatment has a better tumor inhibition effect.

### 3.7. In Vivo Antitumor Research

#### 3.7.1. In Vivo Evaluation of the Efficacy of Peptide Hydrogels

The tumor-bearing mice used for the in vivo antitumor study were divided randomly into five groups, and groups were administered physiological saline (control group), OE peptide hydrogel loaded with GEM (GEM@H group), OE peptide hydrogel loaded with PTX (PTX@H group), GEM and PTX suspension (GEM+PTX group) or OE peptide hydrogel loaded with PTX and GEM (GEM+PTX@H group). The weight of tumor-bearing mice was monitored, and the tumor volume was measured and calculated on days 0, 1, 4 and 7 after administration. As observed in Figure 3A, the body weight of tumor-bearing mice slightly decreased 1 day after subcutaneous injection with PTX and GEM suspension, and the average weight of tumor-bearing mice dropped from 16.5 g to 13.7 g in 6 days. The body weight of mice in the GEM@H and PTX@H groups was more stable; within 7 days after administration, the change trend of the body weight of the tumor-bearing mice in the GEM+PTX@H group showed no significant difference from the blank group. 

The tumor volume changes in mice within 7 days after administration are shown in Figure 3B, from which it can be seen that the tumors in the control group grew rapidly, with the tumor volume increasing rapidly from 97.4 mm^3^ to 743.9 mm^3^ within 7 days. In contrast, the tumors in the GEM+PTX@H group showed a more pronounced inhibitory effect after administration than the GEM+PTX group and even tended to become gradually smaller. The tumor was inhibited after 7 days, and the inhibition rate was 90.84%. Figure 3C,D show the tumor weight and volume of tumor-bearing mice. The minimum average weight of the tumors, which was 0.19 g, was observed in the GEM+PTX@H group. Thus, the inhibition of tumors was highest in the GEM+PTX@H group. These results show that when acting as a drug delivery vehicle, the OE peptide hydrogel can release a large amount of GEM and slowly release PTX. Figure 3E shows the results of hematoxylin and eosin (H&E) staining of the heart, liver, spleen, lung and kidney of the tumor-bearing mice. It shows that in the GEM+PTX group, normal tissues were damaged to some extent. Figure 3F shows the TdT-mediated dUTP nick-end labeling (TUNEL) assay results of the tumor sections from each group of mice. It is evident that the largest fluorescent area is found in the GEM+PTX@H group.

In Figure 3A, the results for the body weights of tumor-bearing mice indicate that direct injection of GEM and PTX suspension has a certain degree of toxicity and side effects in the normal tissues and organs of mice. In contrast, the results in the GEM@H and PTX@H groups illustrate that GEM and PTX encapsulated by the OE peptide hydrogel can be released in responsive to conditions around tumor tissue, and notably, the toxic side effects on normal tissues are reduced. Figure 3B reveals the combination of the massive release of GEM encapsulated in the OE peptide hydrogel and the slow release of PTX. This drug-loading system can not only exert the antitumor effect of the drug itself but also slowly release the drug, further prolonging the time of the drug action, accordingly achieving a more effective antitumor effect. Seven days after administration, the tumors were removed from the right armpit of the tumor-bearing mouse, rinsed and dried immediately. It can be concluded from Figure 3C,D that it can enhance efficacy and reduce toxicity, simultaneously prolonging the drug action. These findings prove that it is a reliable drug-carrying material that enhances antitumor efficacy. Figure 3E indicates that the free drug causes systemic toxic effects, while encapsulating the drug in peptide hydrogels reduces the toxic effects of the drug. Figure 3F indicates that this group has the highest number of apoptotic cells, followed by the GEM+PTX group. The H&E staining and TUNEL experiments revealed that the OE peptide hydrogels loaded with GEM and PTX can enhance efficacy, reduce toxic side effects and exert excellent antitumor effects. This result is consistent with the preceding experimental results.

#### 3.7.2. Biodistribution of Peptide Hydrogel Bodies

Figure 4A shows two groups of in vivo fluorescence images composed of three tumor-bearing mice in each group after injection of free Cy5 suspension and Cy5-loaded OE hydrogel. Ninety-six hours after administration, Cy5 fluorescence in the OE-Cy5 group was clearly concentrated around the tumor tissue, while in the free Cy5 group, the fluorescence strength decreased. Figure 4B shows in vitro fluorescence images of five organs and the tumors of the two groups of tumor-bearing mice 96 h after administration. It can be seen that the fluorescence intensity of tumor tissues in the OE-Cy5 group is more visible than that in the Cy5 group 96 h after administration, and there are some fluorescence signals in both the liver and kidney in the Cy5 group. Figure 4A indicates that the hydrogel can retain the encapsulated active drug in the area surrounding the tumor tissue and prolong the duration of action of the drug. Figure 4B indicates that the OE polypeptide hydrogel can retain the drug near the tumor tissues to exert its drug effect, inhibiting tumor growth and metastasis while reducing toxic side effects on normal tissues.

#### 3.7.3. Biocompatibility of OE Peptide Hydrogel In Vivo

Figure 4C shows the images of dorsal skin tissues stained by H&E after hypodermic injection of the physiological saline or the blank OE polypeptide hydrogel into the backs of mice. As observed in Figure 4C, there is no damage, significant hyperplasia or pathological changes in the dorsal skin tissue of either group, which reveals that the OE polypeptide hydrogel is innocuous in animals and shows a good biosafety profile.

## 4. Conclusions

In conclusion, our injectable pH-responsive OE peptide hydrogels are designed to encapsulate both GEM and PTX drugs to form a stable two-drug co-delivery system. According to the drug release results in vitro and the treatment results of animal experiments, when the drug-loaded OE peptide hydrogel is injected near the tumor tissue and disrupted because of the tumor microenvironment, both drugs are retained in the tumor tissue for sustained and concentrated release, with the hydrophilic drug GEM being released in large quantities and the hydrophobic drug PTX being released slowly until complete release, which not only prolongs the duration of action of the drug but also further enhances its efficacy. In addition, peptides, as biologically active molecules, are safe in vivo. Next, we need to further investigate whether the release process of the two drugs conforms to a certain mathematical model of drug release and to consider the use of this system in combination with immunotherapy and chemotherapy to enhance the immunotherapy effect while enhancing the efficacy of chemotherapy and providing patients with the longest possible survival time.

## Figures and Tables

**Figure 1 pharmaceutics-14-00652-f001:**
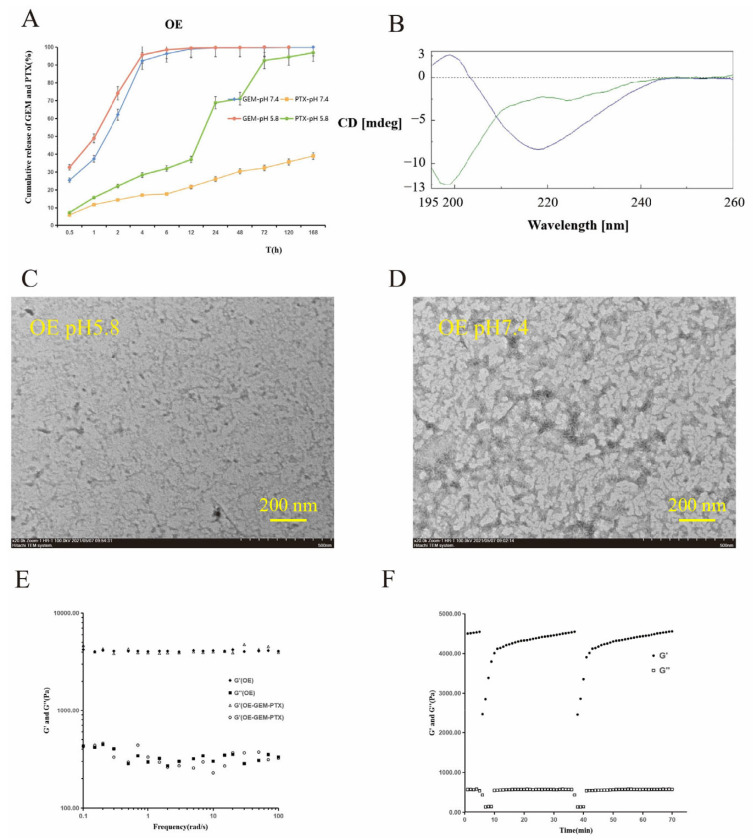
Cumulative release of GEM and PTX from OE hydrogel in different buffer solutions with pH 5.8 and pH 7.4 (**A**). Circular dichroism chromatograms of blank OE peptide in pH 5.8 and pH 7.4 buffers (**B**). Transmission electron microscopy of blank OE peptide at pH 5.8 (**C**) and pH 7.4 (**D**). Dynamic frequency scanning of blank OE hydrogels and GEM+PTX-loaded hydrogels (**E**). Dynamic time scanning of OE peptide hydrogels (**F**).

**Figure 2 pharmaceutics-14-00652-f002:**
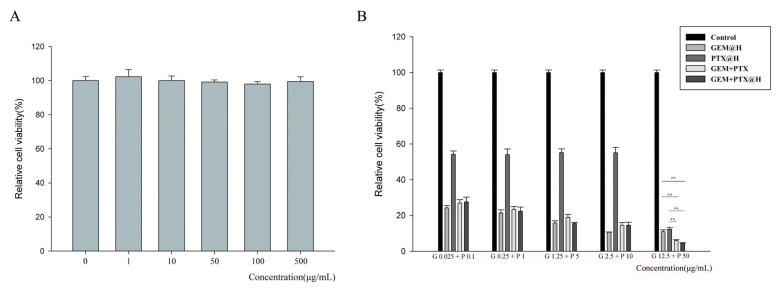
Cell viability of blank OE hydrogel (**A**). The cell inhibition rate of GEM-loaded OE peptide hydrogel, PTX-loaded OE peptide hydrogel, GEM+PTX-loaded OE peptide hydrogel and free GEM+PTX (**B**). ** *p* < 0.01.

**Figure 3 pharmaceutics-14-00652-f003:**
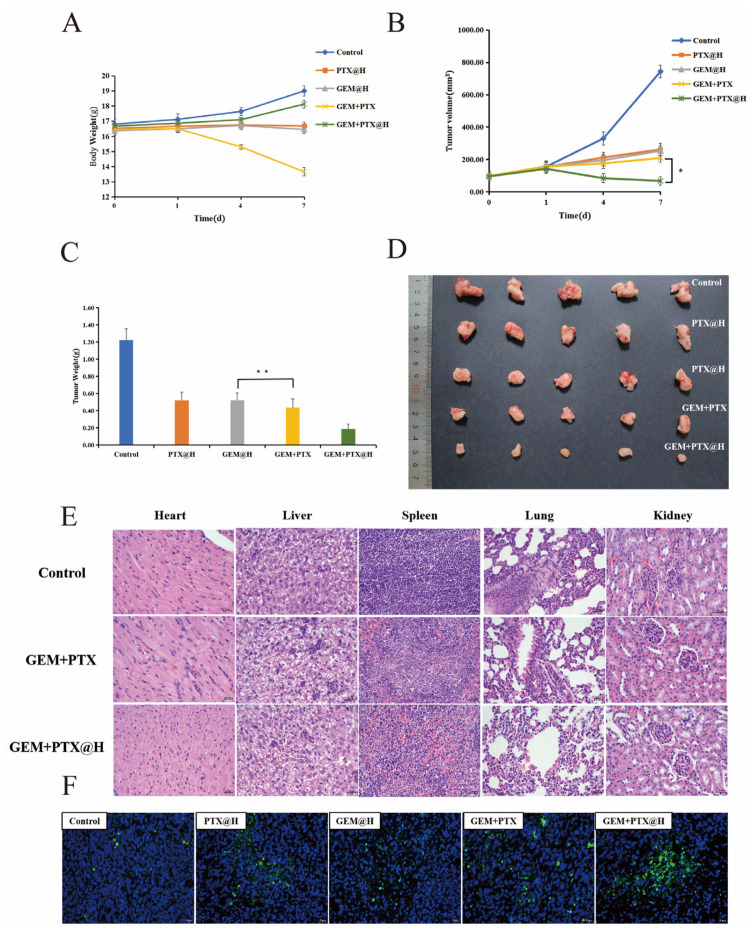
The body weight change curve of mice after administration (**A**). The tumor volume change curve of mice after administration (**B**). The tumor weight of mice 7 days after administration (**C**). Images of the tumors harvested from mice in the control group, the PTX@H group, the GEM@H group, the GEM+PTX group and the GEM+PTX@H group 7 days after administration (**D**). H&E-based immunohistochemical images of five major organs harvested from the tumor-bearing mice 7 days after administration (**E**). TUNEL-based immunohistochemical images of the tumor harvested from a tumor-bearing mouse 7 days after administration (**F**). * *p* < 0.1, ** *p* < 0.01.

**Figure 4 pharmaceutics-14-00652-f004:**
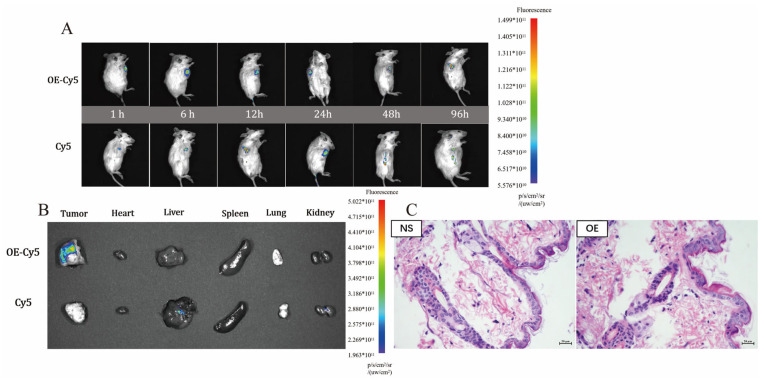
In vivo fluorescence images of tumor-bearing mice over time after administration (**A**), and ex vivo fluorescence images of five major organs harvested from the tumor-bearing mice (**B**) in the free Cy5 and OE-Cy5 groups. H&E-based immunohistochemical images of skin tissue after subcutaneous injection of normal saline and blank OE peptide hydrogel (**C**).

**Table 1 pharmaceutics-14-00652-t001:** Gelation of OE peptides at different concentrations.

Concentration of Peptide (mg/mL)	Result
10	UG
15	SG: 30 s
20	SG: immediately

SG: The time required for stable gel formation; UG: unable to form a stable gel.

**Table 2 pharmaceutics-14-00652-t002:** Gelation of OE peptides at different pH.

pH	Result
6.0	UG
7.4	SG: 30 s
8.0	UG

SG: The time required for stable gel formation; UG: unable to form a stable gel.

## Data Availability

Not applicable.

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
