# Peer review of "pH-Sensitive Peptide Hydrogels as a Combination Drug Delivery System for Cancer Treatment"

_pharmaceutics, 2022, doi:10.3390/pharmaceutics14030652_

Round 1

Reviewer 1 Report

Dear Authors,

Dear Editor,

I read the manuscript and I am sending you my report.

First of all I found the manuscript suitable for the journal and also, in my opinion, it will be of interest for the reader. Although, before publishing I would have some issues to be recommended in order to improve the manuscript and to increase its final visibility.

  • in the introduction part, there is a part describing stimuli responsive materials (last part of page 4 and the first 2 lines of page 5). Among many others, the electric stimuly responsive materials are neglected (for instance, DOI: 10.3390/pharmaceutics13070957)
  • the pH-sensitivity is important, one parameter is related to the pH at which the release rate is changing radically. So, have to experienced pH dependence for your system at more pHs or only at 5.8 and 7.4?
  • you proposed a specific ratio between the two drugs, can you please fundamentate why you choose this ratio? Based on Fig 1A we can see that GEM release is not really dependent on pH but in the case of PTX a pH dependence is visible. Can you further develop these release - pH dependence.
  • Based on my rapid check on the internet,  a dose-dependence can be observed which can be associated also with the releases. For instance, collagen based supports depending on the final anticancer drug amount can exhibit stronger activity (https://doi.org/10.7785/tcrt.2012.500331). 
  • in general, I would be happy to see a more elaborated discussion section, with references to other works.

Best regards,

Reviewer 1

Author Response

  1. in the introduction part, there is a part describing stimuli responsive materials (last part of page 4 and the first 2 lines of page 5). Among many others, the electric stimuly responsive materials are neglected (for instance, DOI: 10.3390/pharmaceutics13070957)

Response: Thanks for your valuable suggestion. We have added this part in the revised manuscript.

  1. the pH-sensitivity is important, one parameter is related to the pH at which the release rate is changing radically. So, have to experienced pH dependence for your system at more pHs or only at 5.8 and 7.4?

Response: Thanks for your valuable comment. We have mainly used the controlled-release application of anti-tumor. The tumor microenvironment is slightly acidic, so we have only investigated at 5.8 and 7.4 at present. The extension of the later topic will continue to investigate other pHs.

  1. you proposed a specific ratio between the two drugs, can you please fundamentate why you choose this ratio? Based on Fig 1A we can see that GEM release is not really dependent on pH but in the case of PTX a pH dependence is visible. Can you further develop these release - pH dependence.

Response: Thanks for your valuable comment. The proportion of drugs was determined based on the report of the combination of paclitaxel and gemcitabine (doi: 10.1186/s12885-018-4936-y. ) and the gelation situation of our peptide. Because gemcitabine is highly hydrophilic, further protonation is easier to release under slightly acidic conditions, while paclitaxel is the opposite. It is precisely by taking advantage of this feature. We first release a large amount of gemcitabine to reach the treatment window and play a long-term therapeutic effect in the sustained-release paclitaxel.

  1. Based on my rapid check on the internet,  a dose-dependence can be observed which can be associated also with the releases. For instance, collagen based supports depending on the final anticancer drug amount can exhibit stronger activity (https://doi.org/10.7785/tcrt.2012.500331). 

Response: Thanks for your valuable suggestion. We have added this part in the revised manuscript.

  1. in general, I would be happy to see a more elaborated discussion section, with references to other works.

Response: Thanks for your valuable suggestion. We have improved discussion section and added other works.

Reviewer 2 Report

The manuscript entitled “pH-sensitive peptide hydrogels as a combination drug delivery system for cancer treatment” submitted by Liu et al. describes the preparation and characterization of a pH-responsive OE peptide hydrogel as a carrier material for anti-tumor drugs gemcitabine (GEM) and paclitaxel (PTX) and its use as a carrier for combined anti-cancer therapy.

Although this manuscript presents an interesting approach, it remains that there are many gaps in form and content.

For the form:

-The authors must improve the handwriting of the manuscript. I suggest merging the "Results" part and the "Discussion" part to make the manuscript more readable.

-In the "Introduction" part, there are many errors. It is not necessary to add the term "etc".

-The terms "net-work", "bio-compatibility", "pro-longs" must be written correctly and the term "stimulus" must be written in the plural.

-In tables 1 and 2, what the term “UL” means.

-The legends of Figures 1 and 2 are not well described.

-The description of the “Rheological study” is hard to understand. It needs more explanation.

For the scientific background:

-The choice of Gemcitabine is not appropriate for co-delivery in the treatment of cancer as long as there is no difference in release at pH 7.4 and pH 5.8 (see Figure 1A). This lack of difference in release that is not sensitive to pH can alter healthy tissues. Gemcitabine is not a good candidate for drug delivery from the peptide hydrogel.

-The 24-hour incubation time for cytotoxicity tests is too short, especially as in vivo experiments last 7 days. An incubation of 48 to 72 hours is generally required for this assay.

In conclusion, authors must significantly improve their manuscript before being accepted for publication in "Pharmaceutics".

Author Response

Referee: 2

-The authors must improve the handwriting of the manuscript. I suggest merging the "Results" part and the "Discussion" part to make the manuscript more readable.

Response: Thanks for your valuable suggestion. We have improved the handwriting and merged the "Results" part and the "Discussion" part.

-In the "Introduction" part, there are many errors. It is not necessary to add the term "etc".

Response: Thanks for your valuable suggestion. We have deleted the term "etc" and corrected errors.

-The terms "net-work", "bio-compatibility", "pro-longs" must be written correctly and the term "stimulus" must be written in the plural.

Response: Thanks for your valuable suggestion. We have corrected these errors

-In tables 1 and 2, what the term “UL” means.

Response: Thanks for your valuable correction. We have corrected “UL” to “UG”.

-The legends of Figures 1 and 2 are not well described.

Response: Thanks for your valuable comment. We have improved the description of Figures 1 and 2.

-The description of the “Rheological study” is hard to understand. It needs more explanation.

Response: Thanks for your valuable comment. We have revised the content of this part

-The choice of Gemcitabine is not appropriate for co-delivery in the treatment of cancer as long as there is no difference in release at pH 7.4 and pH 5.8 (see Figure 1A). This lack of difference in release that is not sensitive to pH can alter healthy tissues. Gemcitabine is not a good candidate for drug delivery from the peptide hydrogel.

Response: Thanks for your valuable comment. Because gemcitabine is highly hydrophilic, further protonation is easier to release under slightly acidic conditions, while paclitaxel is the opposite. It is precisely by taking advantage of this feature. We first release a large amount of gemcitabine to reach the treatment window and play a long-term therapeutic effect in the sustained-release paclitaxel.

-The 24-hour incubation time for cytotoxicity tests is too short, especially as in vivo experiments last 7 days. An incubation of 48 to 72 hours is generally required for this assay.

Response: Thanks for your valuable comment. We have studied the long-term safety investigation in animal experiments.

Reviewer 3 Report

The authors proposed the design and the synthesis of an injectable pH-responsive OE peptide hydrogels able to encapsulate both GEM and PTX drugs, in order to form a stable two-drug co-delivery system. The manuscript is well organized and presented and the experimental set-up is well planned and performed to achieve the work scope. The results are in agreement with the authors theory. However some aspects should be improved.

The abstract should report some significant experimental data.

The introduction should be reduced in the first part, while it should well explain the novelty of the proposed system respect to the similar already reported in literature.

The quality of the figures must be improved because it is really poor and some of them are unreadable.

The English should be revised in the whole manuscript. There are several typos in the manuscript.

Please check the abbreviation in the whole manuscript.

In my opinion this manuscript should be accepted only after these modifications.

Author Response

Referee: 3

  1. The abstract should report some significant experimental data.

Response: Thanks for your valuable suggestion. We have added some significant experimental data in the abstract

  1. The introduction should be reduced in the first part, while it should well explain the novelty of the proposed system respect to the similar already reported in literature.

Response: Thanks for your valuable suggestion. We have reduced the introduction and explained the novelty of our proposed system

  1. The quality of the figures must be improved because it is really poor and some of them are unreadable.

Response: Thanks for your valuable suggestion. We have improved the quality of some figures in the manuscript. We already provide highly quality figures to Assistant editor.

  1. The English should be revised in the whole manuscript. There are several typos in the manuscript.

Response: Thanks for your valuable suggestion. We have revised the whole manuscript and corrected typo mistakes.

  1. Please check the abbreviation in the whole manuscript.

Response: Thanks for your valuable suggestion. We have checked the abbreviation in the whole manuscript.

Round 2

Reviewer 1 Report

Dear authors,

Dear Editor,

We accept the rebbutal and the submission and found it suitable for publication! Some minor corrections can be made during the proof reading!

Best regards,

R1

Reviewer 2 Report

The authors answered to the qiestions of the reviewers and improved their manuscript.